# Determination of Hypoglycemic, Hypolipidemic and Nephroprotective Effects of *Berberis Calliobotrys* in Alloxan-Induced Diabetic Rats

**DOI:** 10.3390/molecules28083533

**Published:** 2023-04-17

**Authors:** Shahid Rasool, Bassam Al Meslmani, Muaaz Alajlani

**Affiliations:** 1College of Pharmacy, University of Sargodha, Sargodha 40100, Pakistan; 2Institute of Pharmaceutical Technology and Biopharmacy, Department of Chemistry and Pharmacy, Friedrich-Alexander-Universität Erlangen-Nürnberg, Cauer Street 4., 91058 Erlangen, Germany; 3Faculty of Pharmacy, Al-Sham Private University, Al-Tal 5910011, Syria

**Keywords:** hypoglycemic, hypolipidemic, nephroprotective, alloxan

## Abstract

Many plants of the Berberis genus have been reported pharmacologically to possess anti-diabetic potential, and *Berberis calliobotrys* has been found to be an inhibitor of α-glucosidase, α-amylase and tyrosinase. Thus, this study investigated the hypoglycemic effects of *Berberis calliobotrys* methanol extract/fractions using in vitro and In vivo methods. Bovine serum albumin (BSA), BSA–methylglyoxal and BSA–glucose methods were used to assess anti-glycation activity in vitro, while in vivo hypoglycemic effects were determined by oral glucose tolerance test (OGTT). Moreover, the hypolipidemic and nephroprotective effects were studied and phenolics were detected using high performance liquid chromatography (HPLC). In vitro anti-glycation showed a significant reduction in glycated end-products formation at 1, 0.25 and 0.5 mg/mL. In vivo hypoglycemic effects were tested at 200, 400 and 600 mg/kg by measuring blood glucose, insulin, hemoglobin (Hb) and HbA1c. The synergistic effect of extract/fractions (600 mg/kg) with insulin exhibited a pronounced glucose reduction in alloxan diabetic rats. The oral glucose tolerance test (OGTT) demonstrated a decline in glucose concentration. Moreover, extract/fractions (600 mg/kg) exhibited an improved lipid profile, increased Hb, HbA1c levels and body weight for 30 days. Furthermore, diabetic animals significantly exhibited an upsurge in total protein, albumin and globulin levels, along with a significant improvement in urea and creatinine after extract/fractions administration for 42 days. Phytochemistry revealed alkaloids, tannins, glycosides, flavonoids, phenols, terpenoids and saponins. HPLC showed the presence of phenolics in ethyl acetate fraction that could be accountable for pharmacological actions. Therefore, it can be concluded that *Berberis calliobotrys* possesses strong hypoglycemic, hypolipidemic and nephroprotective effects, and could be a potential therapeutic agent for diabetes treatment.

## 1. Introduction

Diabetes mellitus is a chronic disorder as old as mankind and has been accounted to cause substantial danger worldwide in the 21st century [1]. It is the most frequent endocrine disease which is caused by a derangement in carbohydrate, fats and protein metabolism. The characteristic hallmark of diabetes mellitus is hyperglycemia with or without glycosuria [2]. Chronic hyperglycemia is coupled with long-term impairment of kidneys, eyes, heart, and blood vessels, ensuing in various problems [3]. Diabetes is an endemic threat to the public and over the past two decades there has been upsurge in its incidence [4]. According to WHO (2003), approximately 177 million people suffered from diabetes in 2000; nevertheless, in 2030 this figure will reach 360 million [5]. High blood glucose levels are also associated with decreasing levels of high-density lipoprotein cholesterol and escalation of low-density lipoprotein cholesterol, thus increasing the risk of coronary heart disease [6]. Therefore, it is vital to manage both diabetes and lipid levels [7].

Currently, diabetes is managed by modification of lifestyle, exercise, diet and long-term use of hypoglycemics or insulin therapy [8]; however, owing to serious side effects of these synthetic drugs (gastrointestinal side effects by metformin and alpha-glucosidase inhibitors, significant hypoglycemia/weight gain by sulfonylureas, frequent dosing, expense and hypoglycemia by meglitinide); there is still a challenge to control diabetes [9]. Despite incredible advancements in the present era, there are insufficient anti-diabetic therapies. Therefore, phytomedicines gain importance, and the search for new lead compounds from plant sources has continued. Scientists are presently focusing on safe and affordable plant-based hypoglycemic agents [10]. From the ethnobotanical data, it has been revealed that more than 800 medicinal plants in Pakistan possess hypoglycemic effects owing to the occurrence in them of flavonoids, glycosides, alkaloids, carotenoids and terpenoids [11]. Plants of the Berberis genus have been used in folkloric medicines in Asia and Europe and their different reported pharmacological effects (including anti-fungal, anti-bacterial, anti-viral, anti-tumor, anti-diabetic and positive effects on cardiovascular and immune system) have been accredited to the existence of important secondary metabolites such as phenolics, alkaloids, tannins, oleanolic acid and many others [12]. One of the important plants of this species, *Berberis calliobotrys* has been found to be an inhibitor of α-glucosidase, α-amylase and tyrosinase [13]. The plant also exhibited anti-convulsant, anti-microbial and anti-oxidant effects. Moreover, essential oils, limonene, ocimene and *p*-cymene as major compounds were found in it through use of gas chromatography [14]. Hence, based on the data already available on the anti-diabetic effects of various Berberis species, and the inhibitory activity of *Berberis calliobotrys* on α-glucosidase and α-amylase, this investigation was carried out to appraise the hypoglycemic potential of *Berberis calliobotrys* through in vitro and in vivo experimental protocols.

## 2. Results

Phytochemical screening: The phytoconstituents comprising proteins, carbohydrates, alkaloids, tannins, glycosides, saponins, flavonoids, terpenoids and fats were detected in both extract and fractions of the plants (Table 1). 

HPLC analysis: The significant hypoglycemic, hypolipidemic and anti-glycation potential was shown by ethyl acetate fraction of the plant, so HPLC was performed to discover its phenolic constituents (Table 2 and Figure 1). The compounds detected included Trans-4-hydroxyl,3-methoxy cinnamic acid, quercetin, gallic acid, caffeic acid, vanillic acid, chlorogenic acid and *p*-coumeric acid.

In vitro anti-glycation activity of *B. calliobotrys* extract and its fractions: The results demonstrate that 1, 0.5 and 0.25 mg/mL concentrations of plant extract and its fractions repressed the second step of AGE generation (Table 3). The control sample showed greatly amplified fluorescence strength owing to AGE complex formation, whereas samples incubated with methanol extract and its various fractions exhibited a considerable decrease in fluorescence intensity by BSA-MGO and BSA-Glucose methods that represent glycation inhibitory activity of the plant.

Effect of *B. calliobotrys* on hypoglycemic activity: Time- and dose-dependent hypoglycemic results; at different doses of plant extract and fractions, at different time intervals in fasted normoglycemic along with alloxan-prompted diabetic rats; are depicted in Table 4. Glibenclamide (10 mg/kg) exhibited highly significant (*p* < 0.001) hypoglycemic effects throughout the study period. Likewise, at 600 mg/kg, methanol extract, butanol as well as ethyl acetate fraction, presented highly significant (*p* < 0.001) hypoglycemic results at all intervals in normal rats. Nevertheless, ethyl acetate fraction and aqueous fraction exhibited significant effects at 2 and 24 h and a highly significant effect at other time intervals. Similarly, lower doses of methanol extract along with different fractions also produced a significant (*p* < 0.05) hypoglycemic effect. Moreover, *B. calliobotrys* methanol extract and its fractions showed time- and dose-reliant hypoglycemic effects in alloxan-tempted diabetic rats from 2–24 h post-treatment with different doses (200, 400 and 600 mg/kg). Furthermore, there is no significant change (*p* > 0.05) occurring at 0 h time and in groups treated with 2% gum acacia suspension. Similarly, current findings revealed that plant extract, butanol and ethyl acetate fractions (600 mg/kg) displayed highly significant (*p* < 0.001) hypoglycemic potential at each interval when administered to alloxan-treated diabetic rats. However, the effects produced by aqueous fraction were significant at 2 h and highly significant at other time intervals.

Effect of *B. calliobotrys* on oral glucose tolerance test: The effect of *B. calliobotrys* methanol extract and its fractions on OGTT in rats is illustrated in Table 5. The methanol extract and n-butanol fraction at 600 and 400 mg/kg unveiled highly significant (*p* < 0.001) hypoglycemic activity until 180 min. Meanwhile, ethyl acetate fraction at 600 mg/kg exhibited a highly significant (*p* < 0.001) decline in blood glucose. Aqueous fraction produced a considerable decline but glucose lowering action was much less than with the standard drug, glibenclamide, which produced remarkably significant blood glucose mitigation (*p* < 0.001) up to 180 min time interval.

Effect of *B. calliobotrys* with or without insulin in alloxanised rats: Methanol extract in combination with 3 units of insulin presented very noteworthy (*p* < 0.001) hypoglycemic activity up to 8th hour which was more pronounced than the hypoglycemic effect of insulin. Table 6 showed that n-butanol fraction at 600 mg/kg in combination with 3 units of insulin produced pronounced a synergistic effect (*p* < 0.001) at various intervals. Likewise, ethyl acetate fraction in combination with 3 units of insulin produced a substantial hypoglycemic effect (*p* < 0.001) at various time intervals in comparison to insulin or ethyl acetate fraction alone. Furthermore, aqueous fraction in combination with insulin showed significant (*p* < 0.001) hypoglycemia at various intervals; however, its effect on blood glucose was less pronounced in comparison to insulin.

Effect of *B. calliobotrys* on mean blood glucose levels: Current findings depicted that methanol extract and butanol fraction (600 mg/kg) considerably (*p* < 0.05) decreased mean blood glucose, whereas its aqueous and ethyl acetate fractions displayed a non-significant reduction (*p* > 0.05) but hypoglycemic effect was much more pronounced (*p* < 0.001) after the 20th and 30th day of treatment (Table 7).

Effect of *B. calliobotrys* on body weight: At 600 mg/kg, methanol extract and its n-butanol fraction stabilized the weight to a highly significant (*p* < 0.001) degree, while its aqueous and ethyl acetate fractions (600 mg/kg) demonstrated substantial change (*p* < 0.05) after the 10th day. Moreover, methanol extract and all its other fractions stabilized the weight highly significantly (*p* < 0.001) at the 30th day in all treated groups (Table 7). 

Effect of *B. calliobotrys* on serum lipid profile: In alloxan diabetic rats, serum cholesterol, triglycerides, LDL and VLDL were markedly increased (*p* < 0.001), while HDL was decreased as compared to normal control (Table 8). The extract and all fractions, to a highly significant degree, reduced TC, TG and LDL-cholesterol levels, whereas a highly significant upsurge in HDL-cholesterol levels was exhibited by methanol extract and its butanol fraction on the 30th day of treatment.

Effect of *B. calliobotrys* on biochemical parameters: The present data revealed that extract and all fractions exhibited a notable improvement in Hb levels, along with a noteworthy decrease in HbA1c levels, compared with the diabetic control group (Table 9). It also depicts the effect of *B. calliobotrys* on different renal parameters after repeated administration for 42 days. Methanol extract, n-butanol and ethyl acetate fraction showed a highly significant rise (*p* < 0.001) in total protein, albumin and globulin levels, while aqueous fraction produced a considerable increase compared to the diabetic control group. Urea and creatinine levels were elevated in diabetic rats. Methanol extract, butanol and ethyl acetate fraction reduced (*p* < 0.001) serum urea and creatinine levels, in contrast to diabetic control rats (Table 9).

## 3. Material and Methods

Plant collection and extraction: Stem and leaves of *B. calliobotrys* (family: Berberidaceae) were procured from mountainous region of Quetta, Pakistan. The plant was recognized and endorsed by Professor Zaheer Ahmad Khan, Botany Department, GC University, Lahore and Professor Rasool Bakhsh Tareen, Botany Department, University of Baluchistan, Quetta. For further reference, voucher sample (No. 1911) was placed in Sultan Ayoub Herbarium, GC University Lahore.

Crude extract of *B. calliobotrys* was made by cold maceration technique. The powdered plant was dipped in 70:30 mixture of water and methanol for 3 days with stirring at regular intervals at room temperature. Thereafter, the mixture was strained and evaporated via rotary evaporator under 760 mmHg pressure. The plant was soaked and filtered two more times. Activity guided fractionation of plant extract was accomplished employing organic solvents (ethyl acetate and n-butanol). The fractions obtained were ethyl acetate, n-butanol and aqueous fractions.

Drugs and chemicals: Alloxan monohydrate (Sigma-Aldrich Chemical Co., St. Louis, MI, USA), glibenclamide (Sonafi-Aventis, Pvt. Ltd., Maharashtra, India), insulin (Regular Humulin-Lilly, USA), gum acacia (Hi-Media Lab., Kennett Square, PA, USA), Glucose oxidase kits (Optium Xceed, Abbot Lab, Abbott Park, IL, USA) were used.

Experimental animals: Male Sprague-Dawley rats (200–250 g) were selected and provided with commercially available rodent diet and water ad libitum.

Phytochemical screening: A number of reported methods were employed to identify several phytoconstituents in extract and fractions [15].

Analysis of plant fraction by HPLC: The procedure of Sultan et al., 2007 [16], as followed to determine the phenolic compounds present in ethyl acetate fraction of *B. calliobotrys* using HPLC technique. 

In vitro Experiment Protocol:In vitro anti-glycation activity using BSA-Glucose method: The protein glycation inhibition was studied by adopting methodology of Matsuura et al., 2002 [17]. The 2 mL of reaction mixture containing glucose (200 mM), BSA i.e., bovine serum albumin (800 ug/mL) and phosphate buffer saline (1 mL; pH 7.4), with extract (1, 0.5 and 0.25 mg/mL) and without extract, was incubated for 7 days at 37 °C in the presence of sodium azide, NaN_3_ (0.2 g/L). Amino guanidine, AG (1 mg/mL) was employed as reference drug. The fluorescence intensity was noted at excitation (370 nm) and emission (440 nm) wavelengths with Perkin Elmer LS-50B spectrofluorometer. Results were stated as percentage inhibition (1).
(1)Inhibition (%)=[(F0−Ft)/F0]×100 

*F*0 and *Ft* represent fluorescence intensity of control and test sample, respectively [18].

b.In vitro anti-glycation activity using BSA-Methylglyoxal method: The protein glycation inhibition at middle stage was investigated in accordance with the method of Peng et al., 2008 [19]. The 300 μL of BSA (10 mg/mL) was combined with 30 μL of MGO (500 mM) in the presence of 0.2 g/L, NaN_3_. Mixture was incubated at 37 °C, with and without plant extract. AG (1 mg/mL) was used as reference standard. The fluorescence intensity was observed with Perkin Elmer LS-50B spectrophotometer at excitation (370 nm) and emission (420 nm) wavelengths. The percentage inhibition was calculated via Formula (2), mentioned below:

(2)Inhibition (%)=[(F0−Ft)/F0]×100 
where, *F*0 and Ft are fluorescence intensity of control and samples [18].

### 3.1. In Vivo Experiment Protocol

Group I (Control): food ad libitum along with water. Group II (Vehicle): orally administered 2% gum acacia. Group III (Positive control): orally administered glibenclamide at dose of 10 mg/kg. Group IV (Treatment): orally administered methanol extract of *B. calliobotrys* and its fractions at 200 mg/kg/day. Group V (Treatment): orally administered methanol extract of *B. calliobotrys* and its fractions at 400 mg/kg/day. Group VI (Treatment): orally administered methanol extract of *B. calliobotrys* and its fractions at 600 mg/kg/day. 

Induction of diabetes in rats: Diabetes was persuaded in overnight fasted rats by giving single intraperitoneal injection of alloxan (150 mg/kg) freshly prepared in normal saline. Diabetes was affirmed 72 h post alloxan injection via glucometer. Rats were selected for experimentation which have fasting blood glucose level of 250 mg/dl. Blood glucose level was estimated from tail vein before and after administration of extract by glucometer. Blood samples were obtained via cardiac puncture. Effect of extract on lipid profile (TC, LDL-cholesterol, and HDL-cholesterol), RFTs (urea, creatinine and blood urea nitrogen), total protein content (albumin and globulin), Hb and HbA1c levels assessed using commercial kits [20].

### 3.2. Hypoglycemic Activity in Oral Glucose Loaded Rats (OGTT)

This test was initially performed in normal rats to assess hypoglycemic potential of plant extract/fractions after oral glucose load [21]. The overnight fasted rats were randomly alienated into six groups (n = 6) as mentioned below. After 30 min of dosing, rats were fed orally with 2 g/kg glucose, and blood was collected from tail before dosing and later at regular intervals 0, 30, 60, 90, 120 and 150 min after glucose administration. Fasting plasma glucose levels were measured via glucose-peroxide reagent strips [2].

Hypoglycemic activity in normal rats: Hypoglycemic activity of extract/fractions was observed in overnight fasted rats by segregating them into 12 groups (n = 6). The first group was normal control. Group 2 was vehicle control and administered 2% gum acacia solution. Groups 3–11 were administered methanol extract and butanol, ethyl acetate fractions at 200, 400 and 600 mg/kg (suspended in 2% gum acacia solution). Group 12 served as standard group (10 mg/kg glibenclamide). Blood glucose levels were tested at 0, 2, 4, 6, 8 and 24 h, subsequent to blood samples collection from tail vein [22,23].

Hypoglycemic activity in alloxan-induced diabetic rats: The 12 groups (n = 6) of overnight fasted diabetic rats were made. Group 1 was normal control. Group 2 (Vehicle control) received only 2% gum acacia solution orally. Groups 3–11 were diabetic and treated with methanol extract, butanol and ethyl acetate fractions (200, 400 and 600 mg/kg). The standard group, Group 12, was orally administered with 10 mg/kg glibenclamide. The blood glucose levels were analyzed at regular intervals by collecting blood from tail vein [22,23].

Synergistic hypoglycemic activity with or without insulin administration: The plant extract/fractions (suspended in 2% gum acacia) were co-administered with insulin. The overnight fasted alloxan-induced diabetic rats were separated into 9 groups (n = 6). The first group was normal control. Group 2 received 2% gum acacia solution. Groups 3, 4 and 5 were co-administered 2 U/kg of insulin + 600 mg/kg of plant extract and butanol, ethyl acetate and aqueous fractions. Group 6, 7 and 8 were administered 3 U/kg of insulin + 600 mg/kg of plant extract and butanol, ethyl acetate and aqueous fractions. Group 9 was given 6 U/kg of insulin alone. Blood glycemic levels were assessed at 0, 2, 4, 6 and 8 h after collecting blood samples from tail vein [23].

Hypoglycemic effect on mean blood glucose level and body weight in alloxan-induced diabetic rats: Mean blood glucose lowering and weight stabilizing effect of *B. calliobotrys* was studied by continuous administration of extract and its fractions for 30 days. Group 1 was normal control. Group 2 was diabetic control. Groups 3, 4 and 5 were orally treated with 600 mg/kg of methanol extract and its fractions. The mean blood glucose level was calculated by collecting blood samples at start (0 day) and then continuously at regular intervals of 10th, 20th and 30th day of experiment, and body weight was continuously analyzed [24].

Hypolipidemic and nephroprotective activity: These activities were determined by continuous administration of extract and fractions for 1 month. Groups 1 and 2 served as normal and diabetes control groups, respectively, while Groups 3, 4 and 5 were orally administered with 600 mg/kg of plant extract/fractions, respectively [25].

### 3.3. Statistical Analysis 

The benefits of multiple comparison were considered optimal in animal testing [26]. The data were stated as mean ± SEM, evaluated by one-way ANOVA followed by Dunnett’s test. *p* < 0.05 was deliberated as statistically significant.

## 4. Discussion

Diabetes has been accepted as a universal health problem with high incidence and mortality [5]. Many years ago, diabetes was managed with a number of medicinal plants and their extracts [27]. However, insulin substitutes from these plant sources should be scientifically evaluated. On account of these reasons, the present study was conducted to validate traditional claims and the active hypoglycemic potential of *Berberis calliobotrys*. Oral administration of plant extract/fractions in alloxanised diabetic rats, as well as in normoglycemic rats, led to insignificant hypoglycemia and improved serum creatinine, urea and blood lipid profile. These observations may be attributed to the nature of different biologically active components, including phenols present in *B. calliobotrys,* as evident from phytochemical screening, as well as HPLC analysis. The hypoglycemic activity in alloxan-persuaded diabetic rats might be owing to the presence of an insulin-like active principle, an alkaloid berberine. Numerous clinical reports on the hypoglycemic action of berberine are present in the literature [28]. It is claimed that it has comparable sugar-reducing potential to sulfonylureas and metformin, but studies are not well designed. It probably acts by stimulating glucose metabolism through the process of glycolysis via increasing glucokinase activity, increased insulin secretion and decreased hepatic gluconeogenesis. It was observed that berberine administration causes a reduction in glucose levels; reduction in HbA1c levels and hypolipidemic action, such as decreased triglycerides and total cholesterol; and low-density lipoproteins [29]. In addition, the hypoglycemic effects of *B. calliobotrys* could be attributed to the phytoconstituents, quercetin, gallic acid and caffeic acid. Ali Asgar, 2013 [30], reported in his study that the above-mentioned compounds possessed anti-diabetic potential. Very recently, a study documented that gallic acid improved insulin sensitivity; reduced obesity, blood pressure and cholesterol levels; however, also tempted adipogenesis in 3T3-L1 adipocytes [31]. Correspondingly, Ahmad et al., 2019 [32], revealed in their study that catechin, ellagic acid and quercetin exhibit anti-oxidant, anti-hyperglycemic, anti-hyperlipidemic and anticancer activity. Qayyum et al., 2016 [33], reported that anti-diabetic and anti-oxidant effects of crude extract of *Heliotropium strigosum* could be possible due to the plant’s phenolic contents, i.e. chromotropic acid, quercetin, trans-4-hydroxy-3-methoxy cinnamic acid, vanillic acid, gallic acid, caffeic acid, m-coumaric acid, *p*-coumaric acid, syringic acid, sinapic acid and ferulic acid. Any antidiabetic agent that is effective in treating diabetes should have the ability to control the increase in glucose level by different mechanisms, so this ability was calculated by oral glucose loaded hyperglycemic models. Methanol extract of *B. calliobotrys* and its fractions exhibited dose-dependent, glucose-reducing effects in glucose loaded hyperglycemic rats. Excessive amounts of glucose in the bloodstream results in more insulin secretion, which stimulates peripheral insulin secretion and increased glucose consumption, and thus controls glucose through a number of mechanisms. However, it is obvious from this study that insulin takes 2–3 h to restore glucose to normal levels. The methanol extract and its butanol and ethyl acetate fractions were found to be more effective in glucose tolerance as compared to the standard drug, glibenclamide. This pronounced action may be attributed to enhanced activity of pancreatic beta cells and excessive insulin secretion. Thus, the mechanism behind this anti-hyperglycemic activity might involve an insulin-like effect. This is probably peripheral glucose consumption, or increased sensitivity of pancreatic beta cells to glucose which results in insulin secretion. Moreover, crude methanol extract and fractions produced pronounced hypoglycemia when co-administered with 3 units of insulin. Previously, similar results were reported on the synergistic hypoglycemic activity of *Conscora deccusata* and *Berberis lycium Royle*, which were administered as an adjunct to various units of insulin [23,34]. Yin et al., 2012 [35], reported that in the presence of insulin, berberine was found to have a synergistic effect on insulin-induced glucose uptake and glucose consumption. The increase in glucose uptake by berberine is of interest because it regulates type 4 glucose transporters (GLUT4), increased glucose metabolism or may increase insulin sensitivity in diabetic rats.

Diabetes mellitus has been perceived to be commonly allied with lipid anomalies such as hypercholesterolemia and hypertriglyceridemia that manifest the marked increase in triglyceride, cholesterol LDL and VLDL along with low concentrations of HDL in serum. These abnormalities in serum lipid profile are caused by disruption of fatty acid metabolism. Additionally, VLDL-transported triglycerides are exchanged with HDL-transported cholesteryl ester, with the aid of CEPT that increases the amount of both atherogenic VLDL remnant particles and triglycerides. Enhanced levels of VLDL-transported triglyceride facilitate the cholesteryl ester transfer protein to enhance conversion of triglyceride to LDL. Moreover, this triglyceride-rich LDL experiences hydrolysis by lipoprotein lipases that results in the formation of lipid-depleted, small, dense LDL particles [36]. The hypolipidemic effects of *Berberis calliobotrys* may be the result of phytoconstituents such as quercetin as this phytochemical is reported to have antihyperlipidemic effects [37]. Another study reported similar findings, showing that *Securigera securidaca* flowers possessed anti-diabetic and anti-hyperlipidemic effects attributed to gallic acid, four hydroxycinnamic acid derivatives (trans-cinnamic, caffeic, chlorogenic and *p*-coumaric acids) [38]. Under normal settings, insulin triggers the lipoprotein lipase enzyme that separates triglycerides; however, in diabetes, lipoprotein lipase is not stimulated because of insulin inadequacy, hence producing hypertriglyceridemia which is a leading cause of cardiac disorders. Thus, the best treatment for diabetes should also employ promising hypolipidemic effects, quite apart from yielding good glycemic regulation [2]. Hence, it could be proposed that *B. calliobotrys* extract could be a potential source of hypolipidemic agents that exert a decline in levels of triglyceride and total cholesterol, and a highly significant improvement in HDL-cholesterol levels, which is desirable to prevent atherosclerosis or ischemic heart disease risk in diabetic patients [39]. The body weight stabilization and decrease in glucose concentration in diabetic rats was also observed in 30 days of treatment with plant extract. 

Diabetes mellitus also results in many complications of the renal system, such as diabetic nephropathy. The outcomes of this study confirmed that the plasma levels of urea and creatinine, major biomarkers of renal dysfunction, were increased in the experimentally induced diabetic rats. But the treatment of diabetic rats with the plant extract reduced the urea and creatinine levels. In the study of Chang et al., 2015 [29], it was examined that berberine significantly decreased the urea and creatinine levels and 24 hr urine protein in streptozotocin-induced diabetic rats. Thus, this implies that *Berberis calliobotrys* can be useful to normalize the function of kidneys in diabetic rats. Moreover, the effect of *B. calliobotrys* on total protein, albumin, globulin, urea and creatinine levels was also determined after 42 days’ continuous treatment with methanol extract and its fractions. The diabetic control rats showed a marked reduction in albumin, globulin and total protein content because of micro proteinuria and albuminuria that are key markers of diabetic nephropathy [40]. In addition, increased level of Hb and decreased glycosylated hemoglobin HbA1c levels were also observed after 42 days’ continuous administration of methanol extract and its tested fractions. HbA1c is a reliable indicator of glycemic control, and decreased HbA1c in diabetes denotes reduced glycation of proteins [41].

In diabetes, glycation is a key molecular-based complication, like chronic hyperglycemia. Glycation is a chemical process involving the binding of the carbonyl group of reducing sugars to amino acids, lipids, proteins and peptides, and enzymatically converting them into AGEs. These AGEs are heterogeneous compounds and present in the form of non-fluorescent cross linking (methylglyoxal-lysine), fluorescent cross linking (pentosidine and non-fluorescent cross linking (CML and pyralline), and are responsible for the development of diabetic complications. In the diabetic condition; a fall in blood glucose is considered as the best approach to limit AGE formation and insulin accumulation and, ultimately; the gluco-regulatory hormone, which increases cell glucose re-uptake, increases glycogenesis and reduces glucagon secretion. A number of phytochemical agents are present in plants, and polyphenols have been found to have anti-glycation activity which acts by mimicking insulin-like activity [42]. The protective effects of *Berberis calliobotrys* against the production of advanced glycation end products (AGEs) formation were assessed via in vitro glycation inhibitory tests. It is noteworthy that plant extract and its tested fractions exhibit more strongly inhibited last stages of glycation.

## 5. Conclusions

Taking in to account the studies discussed above, it is plausible that *Berberis calliobotrys* possesses noteworthy hypoglycemic and hypolipidemic potential, together with substantial glycation-inhibiting activity. These effects could conceivably be owing to the stimulation of insulin discharge from ß-cells and insulin-like principles. Nevertheless, advanced investigations are still essential to segregate its hypoglycemic constituents and determine their exact mode of action and safety.

## Figures and Tables

**Figure 1 molecules-28-03533-f001:**
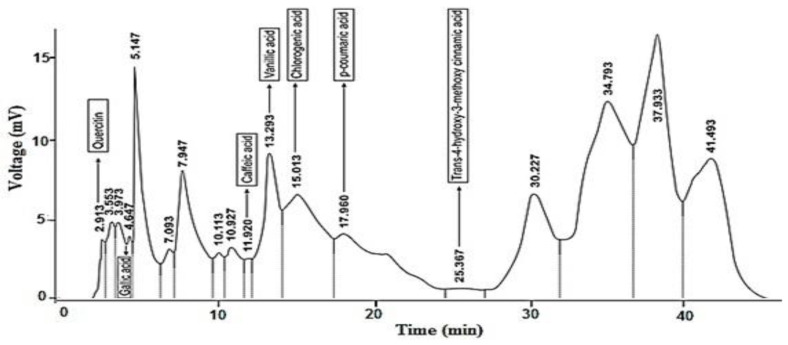
HPLC chromatogram of phenolic components of ethyl acetate fraction of *B. calliobotrys*.

**Table 1 molecules-28-03533-t001:** Phytochemical analysis of *B. calliobotrys*.

Identification Tests	Methanol Extract	Ethyl Acetate Fraction	*n*-Butanol Fraction	Aqueous Fraction
Carbohydrates				
Molisch test	+	-	+	+
Benedict’s test	+	-	-	+
Proteins				
Ninhydrin test	-	-	-	-
Millons test	+	-	+	+
Alkaloids				
Mayer’s test	+	+	+	+
Dragendorff’s test	+	+	+	+
Wagner’s test	+	-	+	+
Hager’s test	+	-	+	+
Glycosides				
Keller-Killiani test	+	+	+	+
Legal’s test	+	-	+	+
Tannins, Phenols				
Ferric chloride test	+	-	+	+
Bromine water test	-	+	+	+
Fats and Fixed Oils				
Spot test	+	-	+	-
Flavonoids				
Lead acetate test	+	-	+	+
Alkaline reagent test	+	-	-	-
Saponins				
Foam test	+	-	+	-
Terpenoids				
Salkowski test	+	+	-	-
Lieber Burchard test	+	-	-	+

+ = Present; - = Absent.

**Table 2 molecules-28-03533-t002:** HPLC profile of ethyl acetate fraction of *B. calliobotrys*.

Sr. No	Compound	Retention Time (min)	Area (mV)	Area (%)	Quantity (ppm)
1	Quercetin	2.913	84.558	0.7	4.45 ± 0.02
2	Gallic acid	4.647	78.887	0.6	2.83 ± 0.04
3	Caffeic acid	11.920	71.175	0.6	3.26 ± 0.03
4	Vanillic acid	13.293	649.745	5.2	40.23 ± 0.03
5	chlorogenic acid	15.013	1069.443	8.6	84.44 ± 0.06
6	*p*-coumeric acid	17.960	1000.535	8.0	13.21 ± 0.04
7	Trans-4-hydroxyl,3-methoxy cinnamic acid	25.367	89.406	0.7	3.11 ± 0.02

**Table 3 molecules-28-03533-t003:** Effect of *B. calliobotrys* on in vitro anti-glycation activity.

Treatment Groups	Conc. (mg/mL)	% Age Inhibition of Protein Glycation	
		BSA-MGO Method	BSA-Glucose Method
Methanol extract	1	81%	91%
	0.5	72%	82%
	0.25	62%	75%
Butanol fraction	1	80%	60%
	0.5	60%	75%
	0.25	57%	68%
Ethyl acetate fraction	1	70%	77%
	0.5	65%	69%
	0.25	52%	60%
Aqueous fraction	1	49%	64%
	0.5	38%	58%
	0.25	31%	47%
Amino guanidine	1	84%	86%
	0.5	76%	76%

**Table 4 molecules-28-03533-t004:** Hypoglycemic effect of *B. calliobotrys* in normal and alloxan-induced diabetic rats.

Treatment Groups	Decrease in Blood Glucose Level in Normal Rats						Decrease in Blood Glucose Level in Diabetic Rats					
	0 h	2 h	4 h	6 h	8 h	24 h	0 h	2 h	4 h	6 h	8 h	24 h
Methanol extract 200 mg/kg	89.0 ± 1.3	85.8 ± 0.2 *	82.0 ± 0.8 **	74.7 ± 0.5 **	72.5 ± 0.9 **	86.8 ± 0.9 ^ns^	438.0 ± 1.9	425.7 ± 1.4 *	391.2 ± 1.0 *	362.5 ± 0.7 **	355.0 ± 2.1 **	421.8 ± 1.0 *
Methanol extract 400 mg/kg	89.3 ± 0.9	83.0 ± 1.5 **	75.7 ± 0.9 **	66.1 ± 0.8 **	62.3 ± 0.3 **	78.2 ± 0.9 **	438.5 ± 2.5	398.0 ± 1.6 **	358.0 ± 1.9 **	302.0 ± 2.7 **	278.0 ± 1.1 **	392.0 ± 2.0 **
Methanol extract 600 mg/kg	90.2 ± 1.3	78.0 ± 0.8 **	66.7 ± 0.8 **	60.7 ± 1.1 **	57.3 ± 0.5 **	74.5 ± 1.5	439.0 ± 1.7	374.5 ± 0.8 **	320.7 ± 3.0 **	285.5 ± 2.4 **	251.8 ± 1.4 **	350.6 ± 1.7 **
Butanol fraction 200 mg/kg	90.0 ± 1.4	86.0 ± 0.4 *	84.3 ± 1.6 *	77.5 ± 1.0 **	75.1 ± 0.8 **	87.8 ± 0.8 ^ns^	436.7 ± 1.5	426.8 ± 0.9 *	404.8 ± 3.2 **	385.0 ± 2.7 **	381.67 ± 1.7 **	430.5 ± 2.2 ^ns^
Butanol fraction 400 mg/kg	90.5 ± 0.8	84.8 ± 0.4 *	75.1 ± 0.6 **	68.7 ± 1.0 **	65.5 ± 0.4 **	84.0 ± 0.4 **	434.0 ± 0.67	401.5 ± 1.0 **	69.7 ± 2.9 **	328.3 ± 1.8 **	309.3 ± 1.5 **	397.0 ± 0.8 **
Butanol fraction 600 mg/kg	91.0 ± 0.6	77.5 ± 1.6 **	70.5 ± 1.3 **	62.8 ± 0.8 **	59.0 ± 0.6 **	79.0 ± 0.5 **	435.7 ± 1.0	376.0 ± 2.2 **	322.0 ± 0.9 **	297.2 ± 2.6 **	277.8 ± 1.3 **	361.9 ± 1.9 **
EA fraction 200 mg/kg	90.3 ± 1.2	88.5 ± 0.7 ^ns^	84.7 ± 0.5 *	78.5 ± 1.3 **	78.0 ± 0.4 **	87.0 ± 0.8 ^ns^	437.0 ± 0.6	434.2 ± 0.7 ^ns^	427.8 ± 1.0 *	418.7 ± 2.3 **	414.0 ± 3.4 **	431.0 ± 1.6 ^ns^
EA fraction 400 mg/kg	89.0 ± 0.4	86.5 ± 0.3 *	79.3 ± 0.4 **	74.5 ± 1.0 **	72.7 ± 2.0 **	84.8 ± 0.3 *	436.0 ± 0.9	427.2 ± 1.4 *	403.67 ± 1.8 **	379.7 ± 1.1 **	371.0 ± 2.8 **	414.2 ± 3.2 **
EA fraction 600 mg/kg	90.0 ± 0.6	84.8 ± 0.9 *	75.3 ± 0.5 **	71.0 ± 1.0 **	67.7 ± 0.9 **	80.5 ± 0.2 **	439.2 ± 3.1	400.0 ± 1.9 **	376.0 ± 2.0 *	340.8 ± 1.6 **	313.2 ± 1.7 **	394.3 ± 0.8 **
Aqueous fraction 200 mg/kg	90.5 ± 1.2	89.0 ± 0.4 ^ns^	87.3 ± 0.7 ^ns^	84.8 ± 0.4 *	84.3 ± 1.1 *	887 ± 0.7 ^ns^	436.3 ± 1.0	435.0 ± 0.9 ^ns^	431.2 ± 1.31 ^ns^	427.7 ± 0.7 *	423.84 ± 0.8 *	435.0 ± 0.8 ^ns^
Aqueous fraction 400 mg/kg	91.3 ± 1.2	87.5 ± 0.6 ^ns^	85.3 ± 0.3 *	81.9 ± 0.7 **	79.5 ± 0.7 **	86.3 ± 0.6 *	438.0 ± 0.9	432.2 ± 0.9 ^ns^	426.0 ± 1.0 *	410.3 ± 1.9 **	399.0 ± 1.5 **	430.5 ± 1.8 *
Aqueous fraction 600 mg/kg	90.0 ± 1.0	84.3 ± 0.4 *	81.5 ± 0.6 **	78.2 ± 0.9 **	75.7 ± 0.6 **	84.0 ± 0.5 *	437.0 ± 1.9	424.2 ± 1.6 *	412.8 ± 2.1 **	391.0 ± 2.2 **	383.5 ± 2.7 **	428.0 ± 1.8 ^ns^
Glibenclamide 10 mg/kg	91.5 ± 0.7	81.3 ± 0.7 **	74.5 ± 0.9 **	70.7 ± 0.4 **	67.8 ± 0.9 **	82.7 ± 0.5 **	437.3 ± 2.21	433.7 ± 1.0 ^ns^	430.8 ± 1.53 *	427.0 ± 0.99 *	428.0 ± 1.8 ^ns^	436.0 ± 2.9 ^ns^
2% Gum Acacia 2 ml/kg	90.0 ± 0.8	89.7 ± 0.4	89.5 ± 1.0	88.3 ± 0.9	88.0 ± 1.3	90.0 ± 0.9	437.6 ± 2.2	437.0 ± 1.0	436.3 ± 1.5	436.0 ± 0.9	435.0 ± 1.8	436.5 ± 2.9

Data are stated as mean ± SEM. *p* < 0.005 was taken as significant compared to vehicle control. ns = non-significant, * = *p* < 0.05, ** = *p* < 0.01. EA = ethyl acetate.

**Table 5 molecules-28-03533-t005:** Effect of *B. calliobotrys* on oral glucose tolerance test.

Treatment Groups	Decrease in Blood Glucose Level						
	0 min	30 min	60 min	90 min	120 min	150 min	180 min
Methanol extract 200 mg/kg	89.6 ± 0.3	106.2 ± 0.9 **	98.7 ± 0.5 *	93.2 ± 0.3 *	89.0 ± 1.0 ^ns^	87.0 ± 1.2 *	84.0 ± 2.8 *
Methanol extract 400 mg/kg	90.2 ± 0.5	96.5 ± 0.4 *	89.3 ± 1.3 ^NS^	85.5 ± 0.9 *	77.3 ± 0.6 **	74.7 ± 0.83 **	70.50 ± 1.26 **
Methanol extract 600 mg/kg	90.00 ± 0.91	86.67 ± 1.04 *	80.33 ± 1.28 **	74.50 ± 0.77 **	70.00 ± 0.7 **	68.0 ± 0.9 **	66.5 ± 0.9 **
Butanol fraction 200 mg/kg	90.5 ± 0.9	112.3 ± 2.5 **	103.2 ± 1.0 **	100.0 ± 0.7 **	96.2 ± 0.8 *	92.8 ± 0.6 ^ns^	89.1 ± 1.0 ^ns^
Butanol fraction 400 mg/kg	89.0 ± 0.3	92.3 ± 0.8 ^ns^	84.7 ± 0.9 *	81.8 ± 0.7 **	77.0 ± 0.5 **	73.2 ± 1.1 **	71.3 ± 0.9 **
Butanol fraction 600 mg/kg	89.0 ± 1.9	87.0 ± 1.0 ^ns^	81.5 ± 0.8 **	77.3 ± 0.9 **	74.8 ± 0.8 **	71.5 ± 0.9 **	69.3 ± 1.2 **
EA fraction 200 mg/kg	89.5 ± 0.3	125.3 ± 0.6 **	117.0 ± 1.5 **	109.7 ± 0.6 **	102.8 ± 0.9 **	96.2 ± 0.4 *	94.5 ± 1.6 *
EA fraction 400 mg/kg	90.0 ± 0.3	121.2 ± 0.5 **	106.0 ± 0.6 **	96.0 ± 1.5 *	94.0 ± 2.3 *	89.7 ± 1.5 ^ns^	85.0 ± 0.7 *
EA fraction 600 mg/kg	89.0 ± 0.7	96.8 ± 1.9 *	90.2 ± 1.4 ^ns^	85.7 ± 0.7 *	83.3 ± 0.8 *	74.0 ± 0.4 **	71.5 ± 1.9 **
Aqueous fraction 200 mg/kg	90.2 ± 1.3	127.5 ± 0.6 **	123.0 ± 0.7 **	114.5 ± 1.0 **	104.3 ± 2.3 **	101.7 ± 1.2 **	97.8 ± 0.9 *
Aqueous fraction 400 mg/kg	90.3 ± 0.6	125.5 ± 0.3 **	117.0 ± 1.0 **	108.7 ± 1.3 **	96.8 ± 0.8 *	93.2 ± 2.6 ^ns^	91.8 ± 1.6 ^ns^
Aqueous fraction 600 mg/kg	89.0 ± 2.6	118.3 ± 0.6 **	107.5 ± 0.5 **	96.8 ± 2.5 *	89.7 ± 1.5 ^ns^	87.3 ± 1.3 ^ns^	84.7 ± 0.4 ^ns^
Glibenclamide 10 mg/kg	89.5 ± 0.7	101.6 ± 0.5 **	95.0 ± 0.4 *	84.5 ± 0.8 *	80.2 ± 0.7 **	78.5 ± 1.3 **	74.8 ± 1.4 **
2% Gum Acacia 2 ml/kg	90.3 ± 0.4	128.7 ± 1.3	125.0 ± 0.8	121.5 ± 0.4	119.3 ± 0.9	116.2 ± 0.8	111.8 ± 0.9

Data are stated as mean ± SEM. *p* < 0.005 was taken as significant compared to vehicle control. ** *p* < 0.01, * *p* < 0.05, ^ns^ = non-significant.

**Table 6 molecules-28-03533-t006:** Synergistic effect of *B. calliobotrys* in diabetic rats, with insulin and without insulin.

Treatment Groups	Decrease in Glucose Level				
	0 h	2 h	4 h	6 h	8 h
Insulin (6 U/kg)	408.3 ± 1.0	293.7 ± 1.5	211.5 ± 1.4	131.8 ± 0.9	102.3 ± 1.2
Insulin (3 U/kg) + methanol extract (600 mg/kg)	404.5 ± 0.9	243.33 ± 1.06 **	178.8 ± 2.3 **	111.8 ± 2.5 **	85.0 ± 2.0 **
Insulin (2 U/kg) + methanol extract (600 mg/kg)	409.0 ± 1.6	258.0 ± 0.9 **	198.1 ± 1.3 **	124.0 ± 1.0 **	94.5 ± 1.2 **
Methanol extract (600 mg/kg)	406.5 ± 2.0	350.3 ± 2.4 **	316.8 ± 1.3 **	274.5 ± 1.46 **	248.0 ± 1.8 **
Insulin 3 U/kg + butanol fraction (600 mg/kg)	410.5 ± 0.9	253.0 ± 1.0 **	194.3 ± 2.0 **	121.8 ± 1.3 **	89.0 ± 1.8 **
Insulin 2 U/kg + BF (600 mg/kg)	409.0 ± 0.7	271.6 ± 0.9 **	208.7 ± 1.9 **	139.8 ± 3.0 **	109.5 ± 2.2 **
Butanolic fraction (600 mg/kg)	407.3 ± 1.3	361.0 ± 2.7 **	314.0 ± 2.1 **	272.0 ± 1.6 **	257.0 ± 0.7 **
Insulin 3 U/kg + EA fraction (600 mg/kg)	403.8 ± 1.2	267.0 ± 1.7 **	217.0 ± 2.0 **	144.0 ± 1.0 **	115.0 ± 0.9 **
Insulin 2 U/kg + EA fraction (600 mg/kg)	404.0 ± 0.8	287.5 ± 0.8 **	229.3 ± 1.0 **	168.5 ± 1.3 **	136.0 ± 1.0 **
EA fraction (600 mg/kg)	409.0 ± 0.5	373.0 ± 0.9 **	347.0 ± 1.6 **	303.0 ± 1.6 **	289.0 ± 1.7 **
Insulin 3 U/kg + Aq fraction (600 mg/kg)	406.5 ± 1.1	300.0 ± 1.0 **	246.0 ± 1.4 **	198.0 ± 1.5 **	173.0 ± 1.0 **
Insulin 2 U/kg + Aq fraction (600 mg/kg)	406.0 ± 1.0	329.0 ± 1.1 **	267.0 ± 1.09 **	213.0 ± 0.50 **	195.0 ± 0.49 **
Aqueous fraction (600 mg/kg)	410.0 ± 0.9	39.0 ± 0.57 **	376.0 ± 1.23 **	351.0 ± 1.45 **	340.0 ± 1.79 **

Data are stated as mean ± SEM. *p* < 0.005 was taken as significant compared to insulin-only group. ** = *p* < 0.01. EA = ethyl acetate, BF = butanol fraction, Aq = aqueous.

**Table 7 molecules-28-03533-t007:** Effect of *B. calliobotrys* on mean blood glucose level and body weight in diabetic rats on 30th day post-treatment.

Treatment Groups	Decrease in Glucose Level(mg/dl)				Change in Body Weight(g)			
	0 Day	10th Day	20th Day	30th Day	0 Day	10th Day	20th Day	30th Day
Methanol Extract 600 mg/kg	426.5 ± 0.9	396.0 ± 1.9 **	354.5 ± 1.2 **	285.8 ± 1.7 **	248.5 ± 1.1	250.5 ± 1.0 ^ns^	251.3 ± 1.5 ns	255.6 ± 2.0 *
Butanol Fraction 600 mg/kg	423.0 ± 1.0	405.3 ± 2.0 **	366.6 ± 1.5 **	297.5 ± 1.1 **	246.0 ± 0.8	246.6 ± 1.0 ^ns^	248.0 ± 1.5 ^ns^	253.3 ± 1.4 *
EA Fraction 600 mg/kg	424.5 ± 1.6	414.8 ± 1.9 *	384.8 ± 2.1 **	313.0 ± 1.0 **	253.0 ± 1.4	248.8 ± 1.6 ^ns^	250.5 ± 1.6 ^ns^	252.0 ± 1.0 ^ns^
Aqueous Fraction 600 mg/kg	428.0 ± 1.88	418.6 ± 1.5 *	407.6 ± 1.3 **	385.8 ± 1.4 **	251.5 ± 1.8	243.3 ± 2.0 **	244.0± 1.3 *	246.5 ± 1.2 ^ns^
Diabetic Control	428.33 ± 1.2	424.0 ± 2.0 ^ns^	421.6 ± 1.1 ^ns^	423.5 ± 1.4 ^ns^	252.0 ± 1.0	239.5 ± 1.4 **	232.33 ± 1.50 **	226.6 ± 1.1 **

Data are stated as mean ± SEM. *p* < 0.005 was taken as significant compared to standard control. ** = *p* < 0.01, * *p* < 0.05, ^ns^ = non-significant. EA = ethyl acetate.

**Table 8 molecules-28-03533-t008:** Effect of *B calliobotrys* on lipid profile in diabetic rats on 30th day post-treatment.

Treatment Groups	Total Cholesterol (mg/dl)	Triglycerides (mg/dl)	HDL- Cholesterol (mg/dl)	LDL- Cholesterol (mg/dl)	VLDL- Cholesterol (mg/dl)
Methanol Extract 600 mg/kg	69.3 ± 0.33 **	66.7± 1.2 **	42.2 ± 0.8 **	38.9± 1.6 **	42.0 ± 1.2 **
Butanol Fraction 600 mg/kg	73.9 ± 1.0 **	71.5± 1.9 **	36.0 ± 0.2 **	47.7 ± 1.6 **	36.0± 1.1 **
EA Fraction 600 mg/kg	85.4 ± 0.5 **	83.7 ± 1.3 **	31.7± 0.3 **	62.2± 0.9 **	31.7 ± 1.0 *
Aqueous Fraction 600 mg/kg	99.8 ± 0.6 **	104.7± 1.3 **	28.3 ± 0.6 **	80.3 ± 0.5 *	24.3 ± 1.0 ^ns^
Normal Control	68.5 ± 0.6	81.2 ± 1.1	46.2 ± 0.8	52.7 ± 0.1	20.2± 1.1
Diabetic Control	108.3 ± 1.2	119.8 ± 1.9	25.3 ± 1.5	86.3 ± 0.9	25.3 ± 1.5

Data are stated as mean ± SEM. *p* < 0.005 was taken as significant compared to diabetic control. * = *p* < 0.05, ** = *p* < 0.01, ^ns^ = non-significant. EA = ethyl acetate.

**Table 9 molecules-28-03533-t009:** Effect of *B. calliobotrys* on biochemical parameters on 42nd day post-treatment.

Treatment Groups	Hb (g/dl)	HbA1c (%)	Serum Urea (mg/dl)	Serum Creatinine (mg/dl)	Total Protein (g/dl)	Albumin (g/dl)	Globulin (g/dl)	A/G Ratio
Normal control	15.00 ± 0.48	4.12 ± 0.67	28.60 ± 1.00	0.84 ± 0.44	6.25 ± 0.56	4.59 ± 0.16	3.76 ± 0.98	1.20 ± 0.14
Diabetic control	8.16 ± 0.44	10.66 ± 0.96	59.3 ± 0.47	1.29 ± 0.28	3.87 ± 1.14	2.27 ± 0.16	2.15 ± 0.67	1.05 ± 0.77
Methanol extract 600 mg/kg	13.67 ± 0.67 **	4.00 ± 0.04 **	32.00 ± 0.67 **	0.68 ± 0.06 **	5.78 ± 0.39 **	4.15 ± 0.12 **	3.50 ± 0.08 **	1.18 ± 0.88 *
Butanol fraction 600 mg/kg	13.00 ± 1.00 **	4.83 ± 0.84 **	38.42 ± 1.24 **	0.74 ± 0.47 **	5.36 ± 0.85 **	3.82 ± 0.24 **	3.29 ± 0.72 **	1.16 ± 1.00 *
EA fraction 600 mg/kg	12.25 ± 0.55 **	5.57 ± 0.23 **	47.09 ± 0.28 **	0.83 ± 0.04 **	4.86 ± 0.04 **	3.30 ± 0.28 **	2.90 ± 0.98 **	1.13 ± 0.47 *
Aqueous fraction 600 mg/kg	10.21 ± 1.16 **	7.78 ± 0.40 **	53.03 ± 1.16 *	0.98 ± 0.55 *	4.43 ± 0.44 **	2.75 ± 0.77 **	2.52 ± 1.47 **	1.09 ± 0.92 *

Data are stated as mean ± SEM. *p* < 0.005 was taken as significant compared to diabetic control. ** *p* < 0.001, * *p* < 0.05. EA = ethyl acetate.

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
