# Peer review of "Determination of Hypoglycemic, Hypolipidemic and Nephroprotective Effects of Berberis Calliobotrys in Alloxan-Induced Diabetic Rats"

_molecules, 2023, doi:10.3390/molecules28083533_

Round 1
Reviewer 1 Report
The manuscript entitled (Determination Of Hypoglycemic, Hypolipidemic And Nephroprotective Effects Of Berberis Calliobotrys In Alloxan H-Induced Diabetic Rats) discussed the benefit role of berberis calliobotrys invivo and in-vitro for their hypoglycemic, hypolipidemic and nephroprotective effects and this is trend subject to avoid the side-effects of the chemical compound. The manuscript is well designed and well written only few corrections needed to be ready for publication.
* Any abbreviations must be mentioned completely at first mention as (hplc) in abstract.
* Science the experiment done on rats with induced dieabits the manuscript should contain ethical statement and approved protocol form the university and lab.
* Number of used animals in materials and methods with detailed information, age, sex, weight, acclomization, rearing....................
* The footnotes of tables of tables from table 4,...... not informative, the table must be understood with the direction of comparison (row, column, within time,.....)
* Some minor languish editing needed for the manuscript.
Author Response
Reviewer 1
The manuscript entitled (Determination Of Hypoglycemic, Hypolipidemic And Nephroprotective Effects Of Berberis Calliobotrys In Alloxan H-Induced Diabetic Rats) discussed the benefit role of berberis calliobotrys invivo and in-vitro for their hypoglycemic, hypolipidemic and nephroprotective effects and this is trend subject to avoid the side-effects of the chemical compound. The manuscript is well designed and well written only few corrections needed to be ready for publication.
Thank you very much for reviewing our manuscript (Determination Of Hypoglycemic, Hypolipidemic And Nephroprotective Effects Of Berberis Calliobotrys In Alloxan H-Induced Diabetic Rats) and for your positive impressions.
* Any abbreviations must be mentioned completely at first mention as (hplc) in abstract.
All abbreviations are mentioned in detail throughout the manuscript.
* Science the experiment done on rats with induced dieabits the manuscript should contain ethical statement and approved protocol form the university and lab.
The approved protocol including the ethical statement are added to the manuscript in accordance with institutional and publisher standards and instructions.
Please refer to line number rang 474
Ethical approval: All applicable international, national, and institutional guidelines for the care were followed. The ethical aspects of the study were critically evaluated by the research ethic committee (REC) of University of Sargodha, Sargodha, Pakistan.
* Number of used animals in materials and methods with detailed information, age, sex, weight, rearing....................
Animals details are highlighted:
Please refer to line number rang 103
Experimental animals: Male Sprague-Dawley rats (200-250 g) were selected and provided with a commercially available rodent diet and water ad libitum.
The number of rats used is elaborated into different subsections in groups for easier reading
* The footnotes of tables of tables from table 4,...... not informative, the table must be understood with the direction of comparison (row, column, within time,.....)
Table 4 footnote has been altered to provide clearer information about table 4 content.
We will work with the publisher to ensure the best shape and arrangement for the final editing of this manuscript.
* Some minor languish editing needed for the manuscript.
The whole manuscript has been checked for any English errors
Thank you once again for reviewing our manuscript and for your time and very useful comments.
Authors
Reviewer 2 Report
Comments to Authors
Authors have presented good work in this research work. All the sections of the articles seems complete, while studying this article I realized it will be better if authors provide equation no to the formula presented in the article like line no 119 in pg 3 and so on. While performing statistical results through ANOVA analysis I suggest authors to provide brief information on the working of ANOVA, why to choose this particular methodology what are the advantage and limitations etc. The rest paper is organized well and results are presented effectively.
Author Response
Reviewer 2
Authors have presented good work in this research work.
Thank you very much for reviewing our manuscript (Determination Of Hypoglycemic, Hypolipidemic And Nephroprotective Effects Of Berberis Calliobotrys In Alloxan H-Induced Diabetic Rats) and for your positive impressions.
All the sections of the articles seems complete, while studying this article I realized it will be better if authors provide equation no to the formula presented in the article like line no 119 in pg 3 and so on.
Equation numbers were added to the article where appropriate.
While performing statistical results through ANOVA analysis I suggest authors to provide brief information, on why to choose this particular methodology what are the advantage .
More detail about statistical analysis methods was elaborated to resolve multiple comparison problems. We have added a reference for more detail.
The rest paper is organized well and results are presented effectively.
We are glad to receive your comment and approval.
Thank you once again for reviewing our manuscript and for the useful and encouraging comments. we are grateful!
Authors
Reviewer 3 Report
In introduction add information related with berberis calliobotrys
Reference should be as per the required format.
Methodology need to improved and elaborate.
Isolated phytoconstituent must be characterized
Table 4 need to be rearranged
Author Response
Thank you very much for reviewing our manuscript (Determination Of Hypoglycemic, Hypolipidemic, and Nephroprotective Effects Of Berberis Calliobotrys In Alloxan H-Induced Diabetic Rats).
We are grateful for the time and effort spent, and your valuable comments. Each of the comments has been considered and your suggestions have been incorporated.
In introduction add information related with berberis calliobotrys
Additional information related to Berberis are added to the introduction
Reference should be as per the required format.
References are now according to the journal authors instructions. Journal specification regarding numbering system and style.
Methodology need to improved and elaborate.
The methodology is now updated for more elaboration and improvements reference were incorporated where appropriate
Table 4 need to be rearranged
Table 4 is now rearranged for easier reading and understanding. A section has been highlighted this will be also checked on manuscript production to ensure the best shape for tables.
We would like to thank you once again for your kind review and valuable comments.
Authors